# S-1 and 5-Fluorouracil-related adverse events in patients with advanced gastric cancer: A meta-analysis

**Qingqing Hu, Jiajia Xu, Jingshu Ke, Ziye Zhang, Ting Chu** *

Nursing School, Zhejiang Chinese Medical University, Hangzhou, Zhejiang Province, People's Republic of China

* chut@zcmu.edu.cn

**Data Availability Statement:** All relevant data are within the paper and its Supporting information files.

**Funding:** The author(s) received no specific funding for this work.

## Abstract

### Objective

To assess S-1 and 5-fluorouracil (5-FU)-related adverse events in patients with advanced gastric cancer and provide focused health care approaches to improve patient compliance and quality of survival.

### Methods

The PubMed, Web of Science, Medline, Cochrane Library, EMbase, SinoMed, Wan Fang Data, CNKI, and VIP databases were searched, and relevant literature was screened from the database construction date through March 31, 2023. Review Manager 5.4.1 and Stata 12.0 were used to analyze the data and GRADEpro was used to assess the quality of the literature. Relative risk ratio (RR) and a 95% confidence interval (CI) were employed as markers to compare adverse events due to S-1 vs 5-FU.

### Results

Eight randomized controlled trials (RCT) were included, which contained 3,455 patients. The S-1 group had 1,804 patients, and the 5-FU group had 1,651 patients. There were 17 recorded adverse events in total. Stomatitis, hypokalemia, mucosal inflammation, and hypophosphatemia were more common in the 5-FU group than in the S-1 group ($P < 0.001$). No significant difference was observed between S-1 and 5-FU for other adverse events.

### Conclusions

Although both S-1 and 5-FU cause a variety of side effects, 5-FU resulted in a higher incidence of stomatitis, hypokalemia, mucosal inflammation, and hypophosphatemia than S-1. Medical professionals should closely monitor the occurrence of adverse drug events and provide timely, rational guidance and nursing care to improve patient compliance and quality of life.

**Competing interests:** The authors have declared that no competing interests exist.

## Introduction

Gastric cancer is one of the top causes of cancer deaths worldwide, with over one million new cases diagnosed each year [1]. In 2018, stomach cancer was reported to have killed 783,000 people (1 in 12 deaths worldwide) [2]. Two Phase III trials in East Asia, the S-1 Adjuvant Chemotherapy for Gastric Cancer (ACTS-GC) trial and the Capecitabine and oxaliplatin adjuvant chemotherapy for Gastric Cancer trial [3, 4], confirmed the efficacy of chemotherapy in gastric cancer.

Fluoropyrimidines are the most commonly used anti-gastric cancer medications [5]. 5-FU is an intravenous fluorouracil medication, and S-1 is another preferred oral fluoropyrimidine for advanced gastric carcinoma [6]. S-1 has the same anticancer efficacy as 5-FU [7], however, 5-FU is associated with an increased risk of stomatitis and diarrhea [8, 9].

Drug compliance is defined as the extent to which a patient takes medication as prescribed by a healthcare provider [10]. A strong treatment plan, good efficacy, and few adverse events are associated with increased patient compliance and quality of life. However, little has been reported regarding drug adherence in cancer patients [11]. Studies suggest that the adverse effects of chemotherapy are the main cause of decreased cancer patient compliance [12–19]. In particular, those who require long-term medication tend to have lower compliance [20]. Failure to adhere to treatment is also a primary cause of disease recurrence or progression [21]. Therefore, healthcare staff must monitor cancer patient compliance to improve patient quality of life.

This study identified S-1 and 5-FU-related adverse events and carried out a meta-analysis comparing the similarities and differences in adverse events between S-1 and 5-FU. The results provide a basis for logical clinical drug use and may improve nursing efficiency, patient compliance, prognosis, and quality of life.

## Methods

### Study design

This meta-analysis was organized according to the Cochrane Handbook for Systematic Reviews of Interventions Version 6.2 [22]. Literature review and data integration were conducted based on PRISMA (Preferred Reporting Items for Systematic reviews and Meta-Analyses) 2020 [23]. The GRADEpro (https://gradepro.org/) system was used to evaluate evidence quality and recommendation grade. The National Cancer Institute Common Terminology Criteria for Adverse Events (NCI-CTCAE) version 3.0 was used to evaluate and grade adverse events: grade 1 = mild, grade 2 = medium, grade 3 = severe, grade 4 = either life-threatening or debilitating, and grade 5 = death. This meta-analysis is registered in the PROSPERO (No. CRD42023409814).

### Inclusion and exclusion criteria

The inclusion criteria were as follows: (1) advanced gastric cancer; (2) treatment with S-1 or 5-Fu; (3) randomized controlled trial; (4) reporting of adverse events.

The exclusion criteria were as follows: (1) study design types: sham RCT, controlled clinical trial (CCT), review, case report, or meta-analysis; (2) test subjects: animal or in *vitro* experiments; (3) unfinished studies; (4) only abstract/title available; (5) no raw data; (6) withdrawn literature; (7) studies with questionable data; (8) studies with more than 80% similarity to published data.

**Table 1. Keywords and strategies used in the literature retrieval strategy.**

| PubMed | Keywords used in literature retrieval |
|---|---|
| #1 | "Stomach Neoplasms" OR "Neoplasm, Stomach" OR "Stomach Neoplasm" OR "Neoplasms, Stomach" OR "Gastric Neoplasms" OR "Gastric Neoplasm" OR "Neoplasm, Gastric" OR "Neoplasms, Gastric" OR "Cancer of Stomach" OR "Stomach Cancers" OR "Gastric Cancer" OR "Cancer, Gastric" OR "Cancers, Gastric" OR "Gastric Cancers" OR "Stomach Cancer" OR "Cancer, Stomach" OR "Cancers, Stomach" OR "Cancer of the Stomach" OR "Gastric Cancer, Familial Diffuse" |
| #2 | "S 1" OR "S1-tegafur-oxonate combination" OR "S1-fluoropyrimidine oxoonate combination" OR "TS-1 cpd" OR "S-1" OR "S-1 cpd" OR "BMS 247616" OR "BMS247616" |
| #3 | "Fluorouracil" OR "5FU" OR "5-FU" OR "5-Fluorouracil" OR "5 Fluorouracil" OR "Fluoruracil" OR "5-FU Lederle" OR "5 FU Lederle" OR "5-FU Medac" OR "5 FU Medac" OR "5-HU Hexal" OR "5 HU Hexal" OR "Adrucil" OR "Carac" OR "Efudix" OR "Fluoro-Uracile ICN" OR "Fluoro Uracile ICN" OR "Efudex" OR "Fluoroplex" OR "Flurodex" OR "Fluorouracil Mononitrate" OR "Fluorouracil Monopotassium Salt" OR "Fluorouracil Monosodium Salt" OR "Fluorouracil Potassium Salt" OR "Fluorouracil-GRY" OR "Fluorouracil GRY" OR "Fluorouracile Dakota" OR "Dakota, Fluorouracile" OR "Fluorouracilo Ferrer Far" OR "Fluracedyl" OR "Haemato-FU" OR "Haemato FU" OR "Neofluor" OR "Onkofluor" OR "Ribofluor" OR "5-Fluorouracil-Biosyn" OR "5 Fluorouracil Biosyn" |
| Literature Retrieval Strategy | ((#1) AND #2) AND #3 |

## Information sources and search strategy

Using the search terms and strategies in Table 1, relevant literature was retrieved from the PubMed, Web of Science, Medline, the Cochrane Library, EMbase, SinoMed, Wan Fang Data, CNKI, and VIP databases. The retrieval period was from the database construction date to March 31, 2023.

## Data extraction

The collected literature was imported into Endnote X9 [24]. After removing duplicate literature, the remaining literature was screened independently by two reviewers. Disagreement was resolved by discussion or with the assistance of a third reviewer. The literature information included author, title, name of the journal, publication year, page number, trial design, trial object, trial method, intervention measures, trial results, and outcomes.

## Quality assessment

The risk of bias of the included literature was analyzed using the Cochrane risk of bias tool, and its quality was assessed using Review Manager 5.4.1 and GRADEpro.

## Data synthesis

Review Manager 5.4.1 and Stata 12.0 were used to analyze the data via relative risk (RR) and 95% confidence interval (CI). $I^2$ was used to test the heterogeneity of the literature [25]. If the data showed $I^2 < 25\%$ (low heterogeneity), they were analyzed using a fixed effect model. If $I^2 \geq 25\%$ (high heterogeneity), data were analyzed using a random effect model and sensitivity analysis [26]. Sensitivity analysis was carried out using replacement effect model, item-by-item elimination, and subgroup analysis. The Egger's test was used to assess publication bias in the literature, and $P < 0.05$ indicated publication bias [27]. A $p$ value of less than 0.05 was considered as statistical significant.

## Results

### Study selection and description

Fig 1 shows that a total of 5,760 relevant studies were retrieved, 5,752 unqualified studies were discarded, and eight qualifying studies were recovered [28–35], with a final total of 3,455 patients. The S-1 group included 1,804 cases and the 5-FU group had 1,651 cases. Table 2 summarizes the key points of the included literature.

### Study quality

The eight included eligible articles were of good overall quality and had a low risk of bias (Figs 2 and 3). Among these, seven studies [28, 29, 31–35] illustrated the process of randomization and six studies [29–33, 35] indicated allocation concealment. GRADEpro evaluation results showed that three studies [28, 29, 32] had high evidence quality and were given a strong recommendation, three studies [33–35] had intermediate evidence quality and were recommended, one study [31] had low evidence quality and was given a weak recommendation, and evidence quality was very low in one study [30] and it was not recommended (Table 2 and Fig 4).

The results of the analysis of the eight included articles revealed that there were differences in the occurrence of adverse medication events between S-1 and 5-FU (Fig 5). The

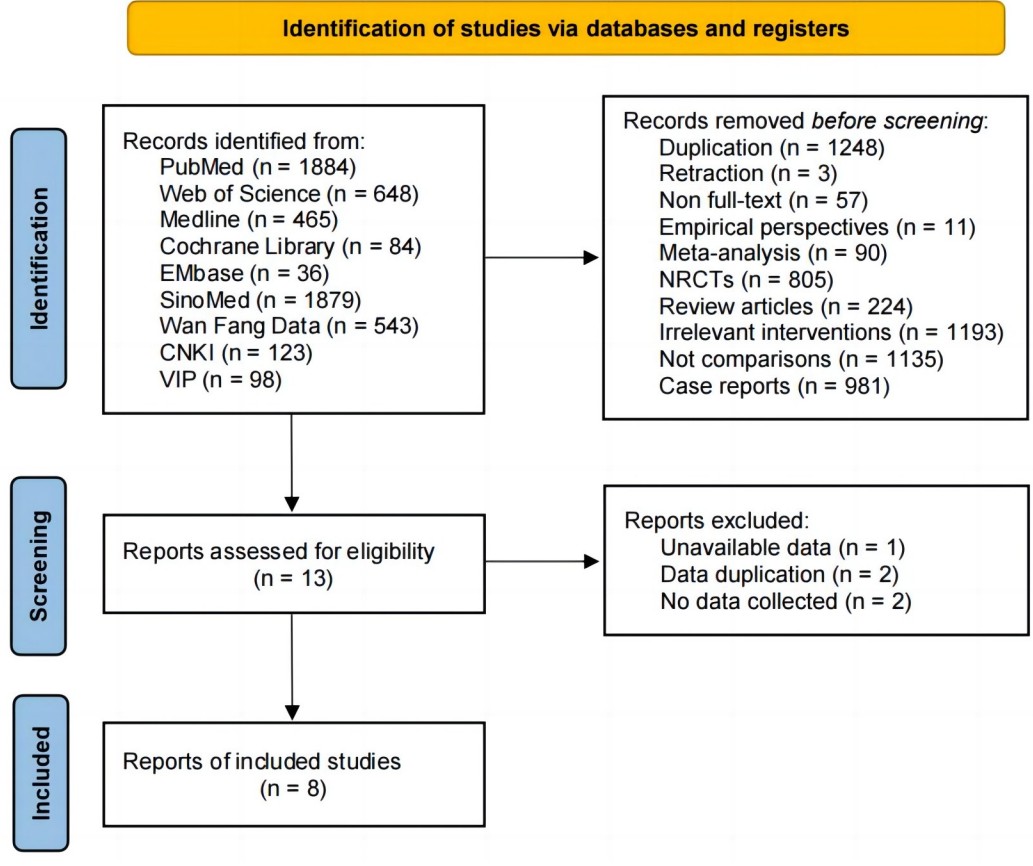

**Fig 1. Flow diagram of the selection process.**

**Table 2. Characteristics of included studies.**

| Study | Country | Experimental group | | | | | Control group | | | | | Certainty of the evidence (GRADEpro) |
|---|---|---|---|---|---|---|---|---|---|---|---|---|
| | | Male (%) | Median age (years) | Method | Agent Exposure | No. | Male (%) | Median age (years) | Method | Agent Exposure | No. | |
| Boku 2009 [28] | Japan | 74.8 | 64.0(58–69) | S-1 (40 mg/m$^2$, bid, days 1–28, every 6 weeks) | median OS 11.4 months (6.4–21.3) | 234 | 75.2 | 63.5(57–69) | 5-FU (800 mg/m$^2$/d, days 1–5, every 4 weeks) | median OS 12.3 months (8.1–19.5) | 234 | High |
| Ajani 2010 [29] | USA | 73.3 | 59.0(18–83) | S-1 (50 mg/m$^2$, bid, days 1–21); cisplatin (75 mg/m$^2$, day 1, every 4 weeks) | median OS 8.6 months | 521 | 68.3 | 60.0(20–85) | 5-FU (1,000 mg/m$^2$/d, 120 h); cisplatin (100 mg/m$^2$, day 1, every 4 weeks) | median OS 7.9 months | 508 | High |
| Lee 2012 [30] | South Korea | 35.0 | 52.0(26–72) | S-1 (80 mg/m$^2$, bid, days 1–14); cisplatin (60 mg/m$^2$, day 1, every 3 weeks) | median cycles 7 (1–8) | 20 | 66.6 | 59.0(28–72) | 5-FU (1000 mg/m$^2$/d, 72 h); cisplatin (80 mg/m$^2$, day 1, every 4 weeks) | median cycles 6 (1–6) | 21 | Very low |
| Nishikawa 2012 [31] | Japan | 70.0 | 68.0(51–81) | S-1 (80 mg/m$^2$/d, days 1–28, every 6 weeks); paclitaxel (80 mg/m$^2$, days 1, 8, 15, every 4 weeks) | median cycles 4 (1–30) | 40 | 65.8 | 67.0(48–79) | 5-FU (800 mg/m$^2$/d, days 1–5, every 4 weeks); paclitaxel (80 mg/m$^2$, days 1, 8, 15, every 4 weeks) | median cycles 3 (1–8) | 38 | Low |
| Ajani 2013 [32] | USA | 73.3 | 59.0(18–83) | S-1 (25 mg/m$^2$, bid, days 1–21); cisplatin (75 mg/m$^2$, day 1, every 4 weeks) | median OS 8.6 months (7.9–9.5) | 521 | 68.3 | 60.0(20–85) | 5-FU(1000 mg/m$^2$/d, days 1–5); cisplatin (100 mg/m$^2$, day 1, every 4 weeks) | median OS 7.9 months (7.2–8.5) | 508 | High |
| Huang 2013 [33] | China | 74.8 | 56.0(18–74) | S-1 (at a body surface area-dependant dosage, 80–120 mg/d, bid, days 1–14); paclitaxel (60 mg/m$^2$, days 1, 8, 15, every 4 weeks) | median time 99 days | 119 | 69.1 | 54.0(19–72) | 5-FU(500 mg/m$^2$, days 1–5); leucovorin (20 mg/m$^2$, days 1–5); paclitaxel (60 mg/m$^2$, days 1, 8, 15, every 4 weeks) | median time 61 days | 110 | Moderate |
| Li 2015 [34] | China | 70.0 | 53.3(41–65) | S-1 (40mg/m$^2$, bid, days 1–21); cisplatin (20mg/m$^2$, days 1–4, every 5 weeks) | median OS 10.00 months (8.59–14.52) | 120 | 73.3 | 55.3(44–66) | 5-FU (800 mg/m$^2$/d, 120 h); cisplatin (20mg/m$^2$, day 1–4, every 4 weeks) | median OS 10.46 months (8.92–13.84) | 116 | Moderate |
| Ajani 2017 [35] | USA | 51.9 | 56.0(25–86) | S-1 (25 mg/m$^2$, bid, days 1–21); cisplatin (75 mg/m$^2$, day 1, every 4 weeks) | median OS 7.5 months (6.7–9.3) | 239 | 49.2 | 56.0(27–83) | 5-FU (800 mg/m$^2$/d, days 1–5); cisplatin (80 mg/m$^2$, day 1, every 3 weeks) | median OS 6.6 months (5.7–8.1) | 122 | Moderate |

No. = Number of patients.

above results were not altered once the included studies were eliminated one by one, or subgroup analysis was done (Fig 6), demonstrating that the results were stable and reliable.

## Meta-analysis of 17 adverse events

Eight studies reported 17 adverse events; details are shown in Table 3. Among these, eight studies [28–35] reported neutropenia, leukopenia, nausea, and anorexia; seven studies [28, 29, 31–35] reported diarrhea; six studies reported [29, 30, 32–35] vomiting; six studies [28, 29, 31, 32, 34, 35] reported stomatitis; six studies [29, 30, 32–35] reported anemia; five studies [28, 29,

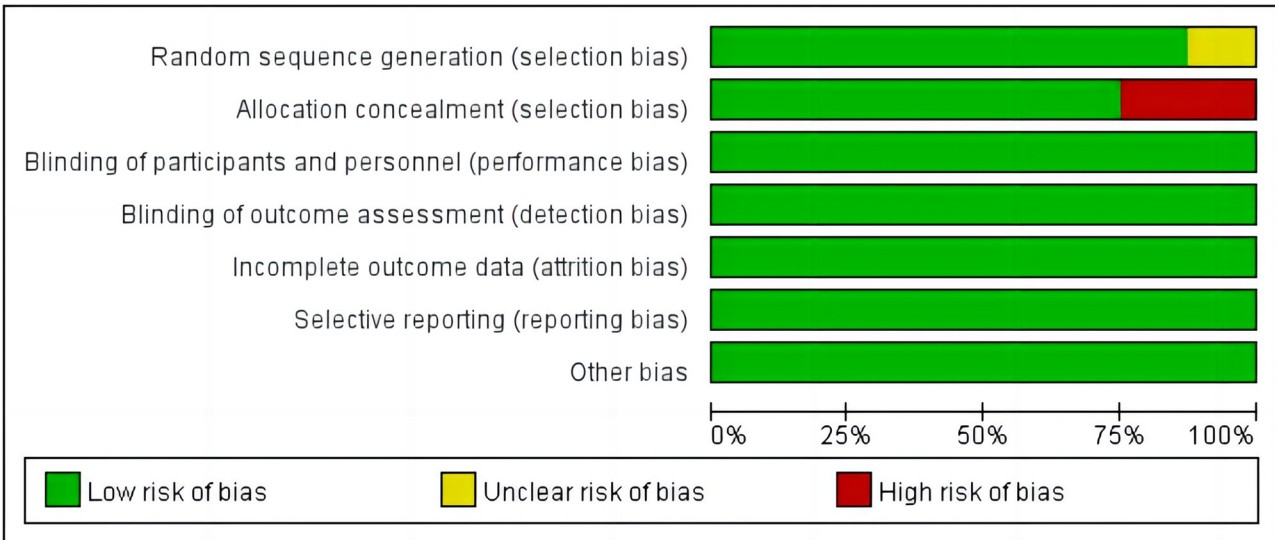

**Fig 2. Risk of bias graph.**

32, 33, 35] reported fatigue; five studies [29, 31, 32, 34, 35] reported thrombocytopenia; four studies [28, 31, 32, 35] reported neuropathy; four studies [29, 31, 32, 35] reported weight loss; four studies [29, 32, 34, 35] reported abdominal pain and hypokalemia; three studies [29, 32, 35] reported mucosal inflammation, hypophosphatemia, and hypomagnesemia. Furthermore, the incidence of stomatitis, hypokalemia, mucosal inflammation, and hypophosphatemia were higher in the 5-FU group than in the S-1 group ($P < 0.001$). For the other adverse events, there was no significant difference between S-1 and 5-FU.

## Meta-analysis of the incidence of stomatitis

As shown in Fig 7, stomatitis was reported in six studies [28, 29, 31, 32, 34, 35], and the heterogeneity test results were: $\chi^2 = 8.92$, $P = 0.11$, $I^2 = 44\%$. The random effects model showed that the incidence in the S-1 group was 1.1% (19/1666) and the 5-FU group was 9.9% (151/1520). The results show that the incidence of stomatitis in the S-1 group was significantly lower than in the 5-FU group ($P < 0.001$).

The results of the subgroup analysis suggested that four studies [29, 32, 34, 35] on S-1+Cisplatin/5-FU+Cisplatin showed that the incidence of stomatitis in the S-1+Cisplatin group was significantly lower than in the 5-FU+Cisplatin group (RR = 0.11, 95% CI [0.06, 0.18], $P < 0.001$). There was only one study focused on S-1+Paclitaxel/5-FU+Paclitaxel and S-1/5-FU individually, but this could not be used for the meta-analysis. Sensitivity analysis of stomatitis was performed item by item, and the results identified one study [28] (Boku N et al.) as the source of heterogeneity.

## Meta-analysis of the incidence of hypokalemia

As shown in Fig 8, hypokalemia was reported in four studies [29, 32, 34, 35], and the heterogeneity test results were: $\chi^2 = 1.07$, $P = 0.78$, $I^2 = 0\%$. Using a fixed effects model, we found that the incidence in the S-1 group was 3.3% (46/1392) and the 5-FU group was 9.4% (118/1250)

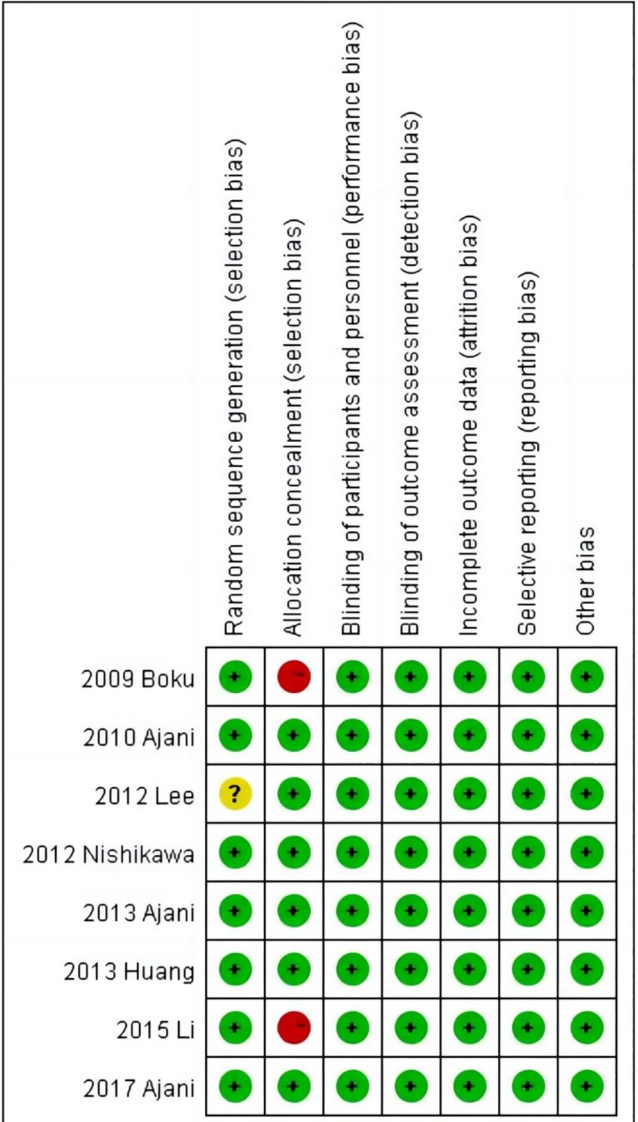

**Fig 3. Risk of bias summary.**

(RR = 0.36, 95% CI [0.25, 0.50], $P < 0.001$). The results showed that the incidence of hypokalemia in the S-1 group was significantly lower than in the 5-FU group.

## Meta-analysis of the incidence of mucosal inflammation

As shown in Fig 9, mucosal inflammation was reported in three studies [29, 32, 35], and the heterogeneity test results were: $\chi^2 = 0.00$, $P = 1.00$, $I^2 = 0\%$. The fixed effects model showed that the incidence in the S-1 group was 0.7% (9/1272) and the 5-FU group was 7.7% (87/1134) (RR = 0.10, 95% CI [0.05, 0.19], $P < 0.001$). This shows that the incidence of mucosal inflammation in the S-1 group was significantly lower than in the 5-FU group.

**S-1 for gastric cancer**

Patient or population: patients with gastric cancer
Settings:
Intervention: S-1

| Outcomes | Illustrative comparative risks* (95% CI) | | Relative effect (95% CI) | No of Participants (studies) | Quality of the evidence (GRADE) | Comments |
|---|---|---|---|---|---|---|
| | Assumed risk Control | Corresponding risk S-1 | | | | |
| 2009 Boku | Study population | | RR 1.36 (1.02 to 1.81) | 466 (1 study) | ⊕⊕⊕⊕ high | |
| | 246 per 1000 | 334 per 1000 (251 to 445) | | | | |
| | Moderate | | | | | |
| | 246 per 1000 | 335 per 1000 (251 to 445) | | | | |
| 2010 Ajani | Study population | | RR 0.6 (0.55 to 0.65) | 1029 (1 study) | ⊕⊕⊕⊕ high | |
| | 890 per 1000 | 534 per 1000 (489 to 578) | | | | |
| | Moderate | | | | | |
| | 890 per 1000 | 534 per 1000 (490 to 578) | | | | |
| 2012 Lee | Study population | | RR 2.21 (0.46 to 10.73) | 40 (1 study) | ⊕⊖⊖⊖ very low[1,2] | |
| | 95 per 1000 | 210 per 1000 (44 to 1000) | | | | |
| | Moderate | | | | | |
| | 95 per 1000 | 210 per 1000 (44 to 1000) | | | | |
| 2012 Nishikawa | Study population | | RR 0.95 (0.47 to 1.93) | 78 (1 study) | ⊕⊕⊖⊖ low[1] | |
| | 289 per 1000 | 275 per 1000 (136 to 559) | | | | |
| | Moderate | | | | | |
| | 290 per 1000 | 275 per 1000 (136 to 560) | | | | |
| 2013 Ajani | Study population | | RR 0.67 (0.6 to 0.75) | 1029 (1 study) | ⊕⊕⊕⊖ moderate | |
| | 699 per 1000 | 468 per 1000 (419 to 524) | | | | |
| | Moderate | | | | | |
| | 699 per 1000 | 468 per 1000 (419 to 524) | | | | |
| 2013 Huang | Study population | | RR 1.85 (1.29 to 2.66) | 229 (1 study) | ⊕⊕⊕⊖ moderate[3] | |
| | 264 per 1000 | 488 per 1000 (340 to 701) | | | | |
| | Moderate | | | | | |
| | 264 per 1000 | 488 per 1000 (341 to 702) | | | | |
| 2015 Li | Study population | | RR 1.59 (1.03 to 2.43) | 236 (1 study) | ⊕⊕⊕⊖ moderate[3] | |
| | 216 per 1000 | 343 per 1000 (222 to 524) | | | | |
| | Moderate | | | | | |
| | 216 per 1000 | 343 per 1000 (222 to 525) | | | | |
| 2017 Ajani | Study population | | RR 1.1 (0.79 to 1.52) | 348 (1 study) | ⊕⊕⊕⊖ moderate[4] | |
| | 305 per 1000 | 336 per 1000 (241 to 464) | | | | |
| | Moderate | | | | | |
| | 305 per 1000 | 335 per 1000 (241 to 464) | | | | |

*The basis for the **assumed risk** (e.g. the median control group risk across studies) is provided in footnotes. The **corresponding risk** (and its 95% confidence interval) is based on the assumed risk in the comparison group and the **relative effect** of the intervention (and its 95% CI).

CI: Confidence interval; **RR:** Risk ratio;

GRADE Working Group grades of evidence
**High quality:** Further research is very unlikely to change our confidence in the estimate of effect.
**Moderate quality:** Further research is likely to have an important impact on our confidence in the estimate of effect and may change the estimate.
**Low quality:** Further research is very likely to have an important impact on our confidence in the estimate of effect and is likely to change the estimate.
**Very low quality:** We are very uncertain about the estimate.

[1] 95 % confidence interval CI crossed and total time number of occurrences or cases did not exceed OIS.
[2] The included RCT studies were all small samples (each group < 30), and the funnel plot was asymmetric.
[3] Total number of occurrences or cases not exceeding OIS.
[4] 95 % confidence interval CI cross line.

**Fig 4. Summary of findings.**

## Meta-analysis of the incidence of hypophosphatemia

As shown in Fig 10, hypophosphatemia was reported in three studies [29, 32, 35], and the heterogeneity test results were: $\chi^2 = 0.99$, $P = 0.61$, $I^2 = 0\%$. The fixed effects model showed that the incidence in the S-1 group was 0.6% (8/1272) and the 5-FU group was 4.3% (49/1134) (RR = 0.14, 95% CI [0.07, 0.31], $P < 0.001$). This shows that the incidence of hypophosphatemia in the S-1 group was significantly lower than in the 5-FU group.

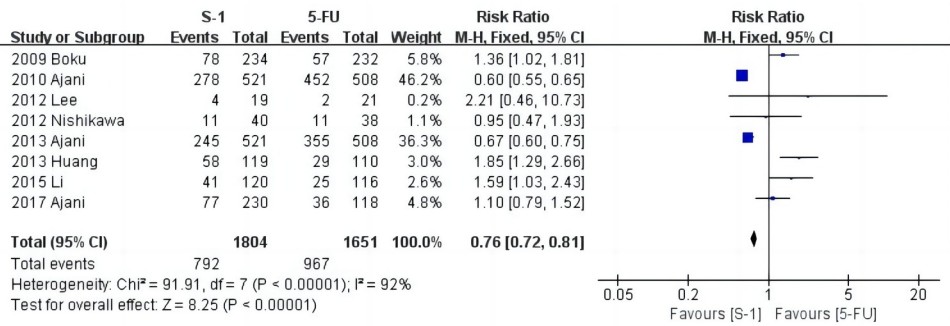

**Fig 5. Forest plot of adverse events.**

## Publication bias

According to the Egger's test results (Table 3 and Fig 11), publication bias existed in the eight included studies. Publication bias was observed for neutropenia, fatigue, neuropathy, weight loss, and abdominal pain ($P < 0.05$); publication bias for mucosal inflammation, hypophosphatemia, and hypomagnesemia was not obtained. Other adverse events showed no publication bias ($P > 0.05$).

## Discussion

### Methodological quality of the included studies

Eight RCTs containing 3,455 subjects were included in this study. There were 1,804 cases in the S-1 group and 1,651 cases in the 5-FU group. The overall quality of the included literature

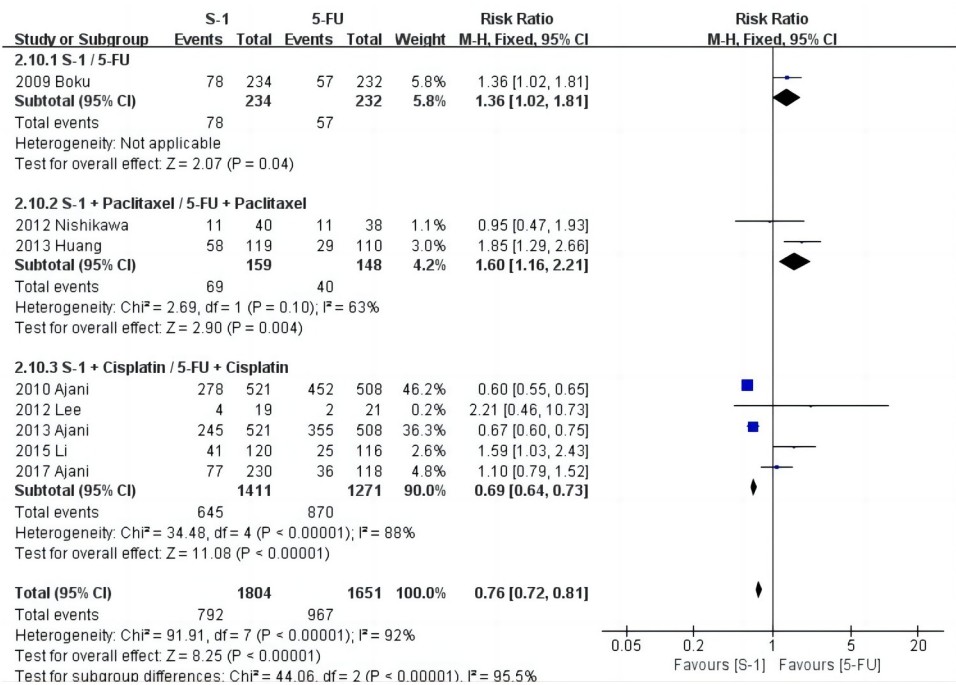

**Fig 6. Forest plot of adverse events (subgroup analysis).**

**Table 3. Adverse events (grade ≥ 3) comparison between S-1 and 5-FU chemotherapy in meta-analysis.**

| Adverse events (grade ≥ 3) | Study counts | S-1 | | | 5-FU | | | | Z | P | Heterogeneity of study design | | | Model | Egger's test | References |
|---|---|---|---|---|---|---|---|---|---|---|---|---|---|---|---|---|
| | | Events | Total | % | Events | Total | % | RR [95% CI] | | | $\chi^2$ | P | $I^2$ | | | |
| Abdominal pain | 4 | 91 | 1392 | 6.5 | 56 | 1250 | 4.5 | 1.49 [1.08,2.07] | 2.41 | 0.02 | 1.96 | 0.58 | 0% | Fixed | 0.044 | [29, 32, 34, 35] |
| Anemia | 6 | 257 | 1530 | 16.8 | 227 | 1381 | 16.4 | 0.95 [0.78,1.15] | 0.54 | 0.59 | 9.45 | 0.09 | 47% | Random | 0.424 | [29, 30, 32–35] |
| Anorexia | 8 | 112 | 1804 | 6.2 | 103 | 1651 | 6.2 | 1.02 [0.79,1.31] | 0.12 | 0.91 | 1.28 | 0.99 | 0% | Fixed | 0.685 | [28–35] |
| Diarrhea | 7 | 86 | 1785 | 4.8 | 57 | 1630 | 3.5 | 1.41 [0.82,2.43] | 1.25 | 0.21 | 10.14 | 0.12 | 41% | Random | 0.208 | [28, 29, 31–35] |
| Fatigue | 5 | 166 | 1625 | 10.2 | 143 | 1476 | 9.7 | 1.24 [0.82,1.88] | 1.02 | 0.31 | 8.59 | 0.07 | 53% | Random | 0.029 | [28, 29, 32, 33, 35] |
| Hypokalemia | 4 | 46 | 1392 | 3.3 | 118 | 1250 | 9.4 | 0.36 [0.25,0.50] | 6.06 | <0.001 | 1.07 | 0.78 | 0% | Fixed | 0.568 | [29, 32, 34, 35] |
| Hypomagnesemia | 3 | 9 | 1272 | 0.7 | 26 | 1134 | 2.3 | 0.32 [0.10,1.01] | 1.95 | 0.05 | 3.15 | 0.21 | 36% | Random | NA | [29, 32, 35] |
| Hypophosphatemia | 3 | 8 | 1272 | 0.6 | 49 | 1134 | 4.3 | 0.14 [0.07,0.31] | 4.97 | <0.001 | 0.99 | 0.61 | 0% | Fixed | NA | [29, 32, 35] |
| Leukopenia | 8 | 196 | 1804 | 10.9 | 270 | 1651 | 16.4 | 1.18 [0.59,2.36] | 0.47 | 0.64 | 57.87 | <0.001 | 88% | Random | 0.084 | [28–35] |
| Mucosal inflammation | 3 | 9 | 1272 | 0.7 | 87 | 1134 | 7.7 | 0.10 [0.05,0.19] | 6.71 | <0.001 | 0.00 | 1.00 | 0% | Fixed | NA | [29, 32, 35] |
| Nausea | 8 | 111 | 1804 | 6.2 | 129 | 1651 | 7.8 | 0.81 [0.63,1.03] | 1.72 | 0.09 | 1.82 | 0.97 | 0% | Fixed | 0.493 | [28–35] |
| Neuropathy | 4 | 7 | 1025 | 0.7 | 4 | 896 | 0.4 | 1.33 [0.45,3.98] | 0.51 | 0.61 | 2.48 | 0.48 | 0% | Fixed | 0.028 | [28, 31, 32, 35] |
| Neutropenia | 8 | 445 | 1804 | 24.7 | 596 | 1651 | 36.1 | 1.31 [0.76,2.24] | 0.98 | 0.33 | 108.29 | <0.001 | 94% | Random | 0.020 | [28–35] |
| Stomatitis | 6 | 19 | 1666 | 1.1 | 151 | 1520 | 9.9 | 0.17 [0.08,0.36] | 4.59 | <0.001 | 8.92 | 0.11 | 44% | Random | 0.361 | [28, 29, 31, 32, 34, 35] |
| Thrombocytopenia | 5 | 98 | 1432 | 6.8 | 117 | 1288 | 9.1 | 1.14 [0.57,2.31] | 0.37 | 0.71 | 14.55 | 0.006 | 73% | Random | 0.103 | [29, 31, 32, 34, 35] |
| Vomiting | 6 | 102 | 1530 | 6.7 | 114 | 1381 | 8.3 | 0.83 [0.64,1.08] | 1.40 | 0.16 | 2.87 | 0.72 | 0% | Fixed | 0.575 | [29, 30, 32–35] |

5-FU = 5-fluorouracil; % = accumulated percentage; RR = risk ratio; CI = confidence interval; NA = not available.

Adverse events (grade≥3) were assessed by the National Cancer Institute Standard for Common Terminology for Adverse Events (NCI-CTCAE) Version 3.0.

was high (there were six studies of high and medium quality in total). In addition, different treatment regimens may have contributed to the heterogeneity.

## Prevention and monitoring of stomatitis

In this study, adverse events were identified as the main cause of poor compliance. In addition, the incidence of stomatitis in the S-1 group was significantly lower than in the 5-FU group ($P < 0.001$). Cancer patients treated with chemotherapy often experience stomatitis, which can lead to dysphagia, altered taste, weight loss, secondary infection, a longer hospital stay, and a lower quality of life [36]. The use of some nursing measures (propolis or cryotherapy) can help prevent stomatitis [37], and zinc chloride and sodium bicarbonate mouthwash can also be used [38]. Therefore, medical workers should monitor the occurrence of stomatitis in

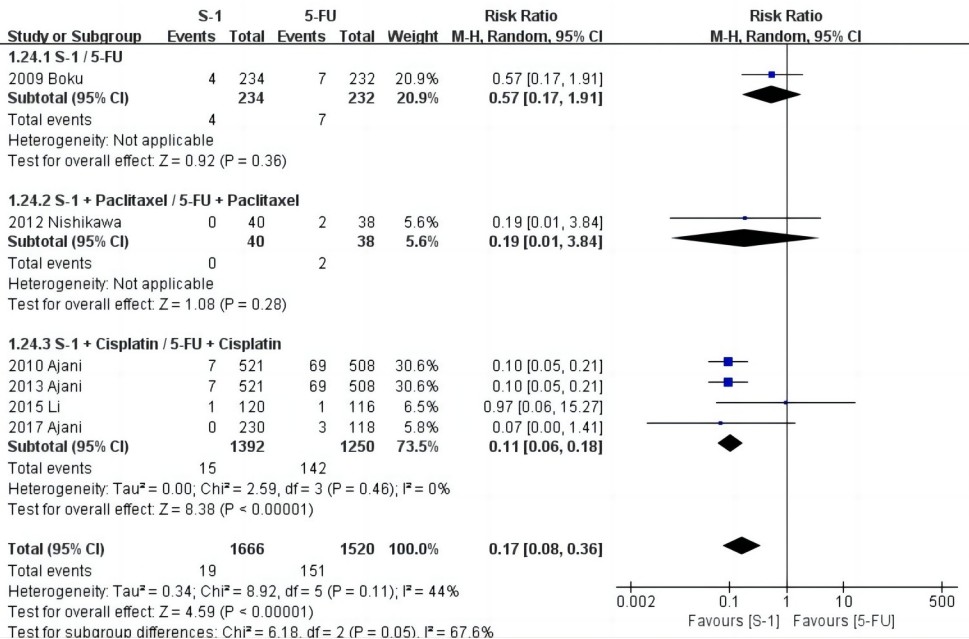

**Fig 7. Forest plot of stomatitis.**

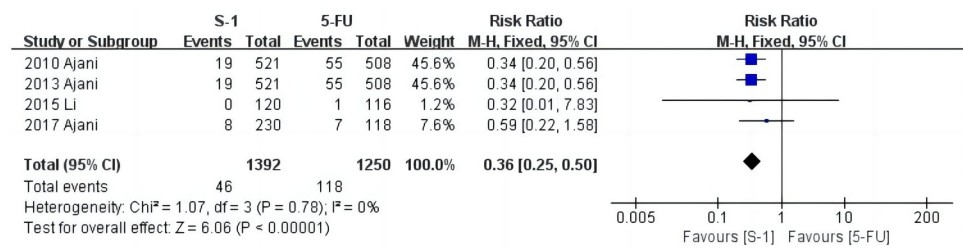

**Fig 8. Forest plot of hypokalemia.**

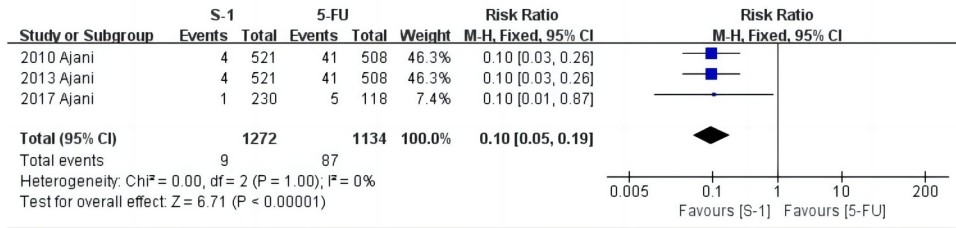

**Fig 9. Forest plot of mucosal inflammation.**

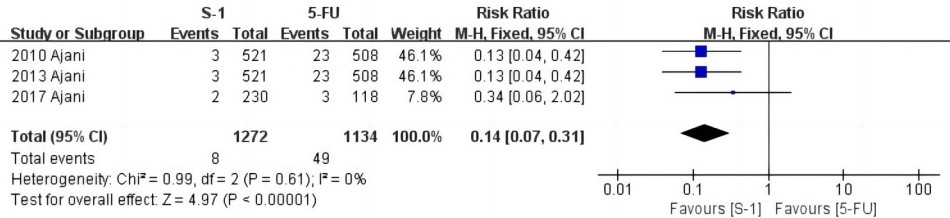

**Fig 10. Forest plot of hypophosphatemia.**

patients with advanced gastric cancer treated with 5-FU, and take proactive steps to prevent and treat it.

## Prevention and monitoring of hypokalemia

The incidence of hypokalemia was lower in the S-1 group than in the 5-FU group ($P < 0.001$). Muscle weakness, paralysis, arrhythmia, paraplegia, and rhabdomyolysis can all result from hypokalemia [39]. Therefore, medical workers should monitor patient lack of strength and muscle pain, and actively prevent and treat hypokalemia.

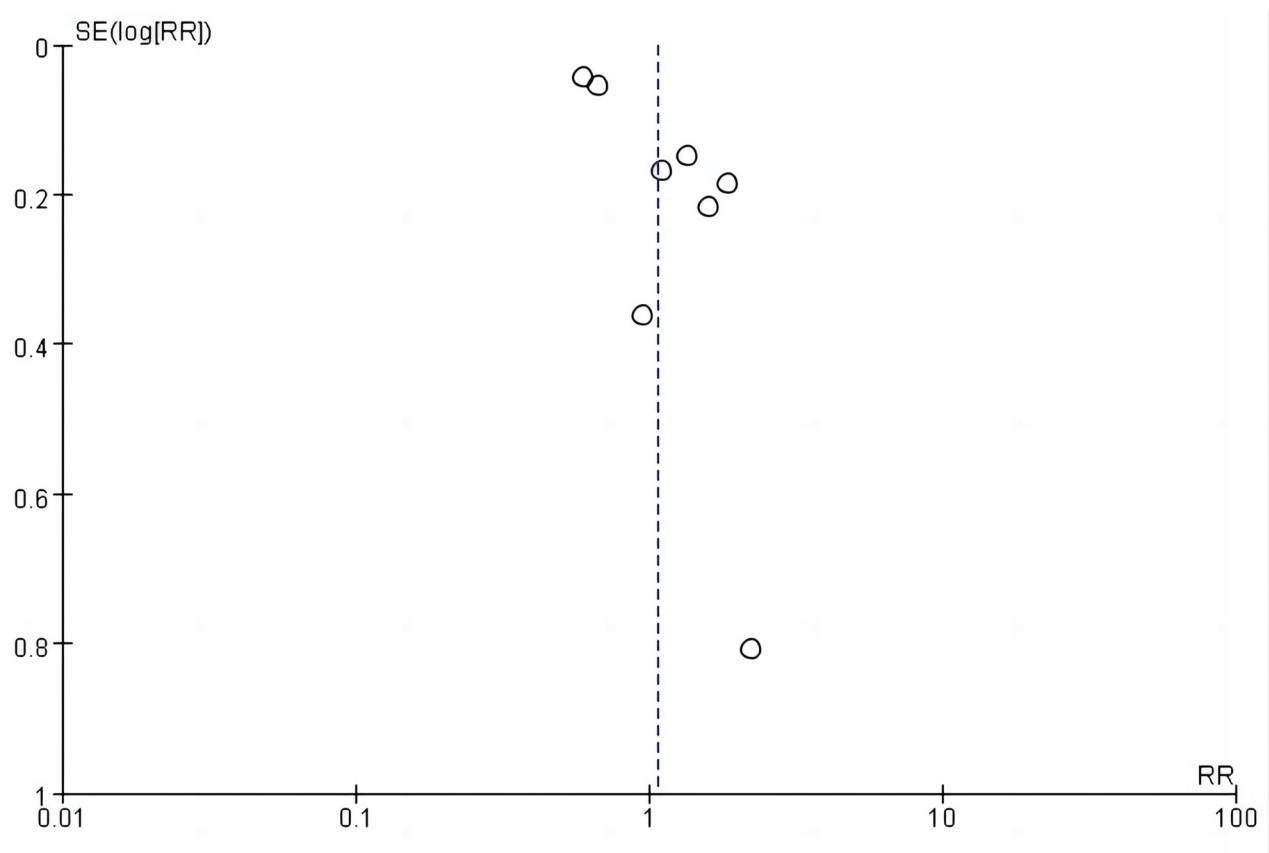

**Fig 11. Funnel plot.**

### Prevention of mucosal inflammation

Mucosal inflammation [40] was significantly less common in the S-1 group than in the 5-FU group ($P < 0.001$). Mucosal inflammation can reduce compliance [41, 42], compromising treatment efficacy and patient prognosis. Oral glutamine, sucrose, and / or trehalose [43] are helpful in preventing mucosal inflammation and improving drug compliance. As a result, medical workers should be alert to the presence of mucosal inflammation in patients and take proactive measures to prevent and treat this condition.

### Monitoring of blood phosphate

When compared to the 5-FU group, the S-1 group had a significantly lower incidence of hypophosphatemia ($P < 0.001$). Hypophosphatemia can trigger drowsiness, dizziness, rhabdomyolysis, impaired bone mineralization, respiratory failure, central nervous system dysfunction, and hemolytic anemia [44, 45]. As a consequence, medical workers should monitor changes in patient blood phosphate levels and aggressively avoid excessive reductions in blood phosphate.

### Oral therapy is more acceptable

Evidence suggests that oncology patients prefer oral therapy over intravenous therapy [46]. In addition, oral therapy promotes treatment convenience while lowering the risk of complications and additional expenditure from intravenous administration [47].

### Implications for nursing practice and further research

Gastric cancer is the fifth most common malignant tumor and the third leading cause of cancer death in the world [2]. This study provides a new foundation for the selection of fluoropyrimidines in patients with advanced gastric cancer. It also suggests new ideas and insights on improving patient compliance in terms of adverse drug events, and experimental evidence on how to improve nursing efficiency, nursing job satisfaction, and patient quality of life. However, present research on chemotherapy compliance in cancer patients and the improvement of linked care measures remains sparse, suggesting this as a potential research topic.

### Strengths

The adverse events following S-1 and 5-FU treatment in patients with advanced gastric cancer were analyzed in this study. The results suggested that both S-1 and 5-FU caused adverse events. Moreover, 5-FU treatment resulted in a higher frequency of stomatitis, hypokalemia, mucosal inflammation, and hypophosphatemia than S-1. Identification of the incidence of adverse events resulting from different therapies can aid in the selection of therapeutic drugs that may achieve better compliance. This will also improve the efficacy of chemotherapy in patients with advanced gastric cancer, improve medical and nursing measures, and improve patient quality of life and satisfaction with nursing care. For this reason, we suggest that S-1 is a better medication regimen than 5-FU in patients with advanced gastric cancer.

### Limitations

The following are the limitations of this study: (1) the insufficient design of the two included RCTs affected the objectivity of the results; (2) the included literature exhibited language bias, and the absence of negative results and grey literature may have adversely affected the results or conclusions. As a result, these findings and conclusions require a future study comprised of large-sample, high-quality RCT investigations for further confirmation and validation.

## Conclusions

The available evidence supports the preferred use of S-1 treatment in patients with advanced gastric cancer. The S-1 regimen less frequently results in stomatitis, hypokalemia, mucosal inflammation, and hypophosphatemia than the 5-FU regimen. Therefore, medical workers should monitor adverse events caused by chemotherapy, formulate corresponding response measures to improve the efficiency and satisfaction of nursing care, and improve patient compliance. These measures will improve patient prognosis and quality of life.

## Supporting information

**S1 Checklist. PRISMA 2020 checklist.**
(DOCX)

## Author Contributions

**Conceptualization:** Ting Chu.

**Data curation:** Jiajia Xu.

**Formal analysis:** Jingshu Ke, Ting Chu.

**Methodology:** Ziye Zhang.

**Supervision:** Ting Chu.

**Validation:** Ting Chu.

**Writing – original draft:** Qingqing Hu.

**Writing – review & editing:** Ting Chu.

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
