## [Decision Letter · Decision Letter 0]

13 Mar 2023

PONE-D-23-03394S-1 and 5-Fluorouracil-related adverse events in patients with advanced gastric cancer: A meta-analysisPLOS ONE

Dear Dr. Chu,

Thank you for submitting your manuscript to PLOS ONE. After careful consideration, we feel that it has merit but does not fully meet PLOS ONE’s publication criteria as it currently stands. Therefore, we invite you to submit a revised version of the manuscript that addresses the points raised during the review process.

We look forward to receiving your revised manuscript.

Kind regards,

Wala BEN KRIDIS, M.D., Ph.D

Academic Editor

PLOS ONE

Journal Requirements:

2. Please note that PLOS ONE has specific guidelines on code sharing for submissions in which author-generated code underpins the findings in the manuscript. In these cases, all author-generated code must be made available without restrictions upon publication of the work. 

Please review our guidelines at https://journals.plos.org/plosone/s/materials-and-software-sharing#loc-sharing-code and ensure that your code is shared in a way that follows best practice and facilitates reproducibility and reuse.

**Additional Editor Comments:**

The manuscript evaluated the adverse effects of two chemotherapy agents, suggesting that S-1 should be preferred in the treatment of gastric cancer.

There are some details in the manuscript that need to revise, such as the literature in the risk assessment map should be arranged by year. In addition, the formatting of the manuscript is very terrible, especially for the layout of figures and tables, which need to be completely revised.

Reviewers' comments:

Reviewer's Responses to Questions

**Comments to the Author**

1. Is the manuscript technically sound, and do the data support the conclusions?

Reviewer #1: Partly

Reviewer #2: Yes

2. Has the statistical analysis been performed appropriately and rigorously? 

Reviewer #1: No

Reviewer #2: I Don't Know

3. Have the authors made all data underlying the findings in their manuscript fully available?

Reviewer #1: No

Reviewer #2: Yes

4. Is the manuscript presented in an intelligible fashion and written in standard English?

Reviewer #1: No

Reviewer #2: Yes

5. Review Comments to the Author

Reviewer #1: Although the study was of great significance to explain the adverse events or reactions of 5-FU and S-1, the methods and reports of the manuscript did not meet the purpose of the study. My comments are as follows.

1. The review only included RCT and other types of studies such as non-randomized trials and case series reports were not collected. According to the PRISMA harms statement, these types of research advantages are important data sources.

2. Only adverse events were reported in the review. Adverse events need to analyze their causality with 5-FU and S-1. More importantly, in addition to the adverse events, these adverse reactions should also be reported and analyzed.

Reviewer #2: The manuscript evaluated the adverse effects of two chemotherapy agents, suggesting that S-1 should be preferred in the treatment of gastric cancer. It has certain significance to guide the clinical use of chemotherapy agents. But, some issues should be mentioned.

1. The retrieval dates of the include literature in the manuscript need to be updated to ensure the rigor of the manuscript and the accuracy of the data included.

2. Has the PROSPERO registration been conducted for this Meta-analysis? If so, please provide the registration number.

3. By reading the entire manuscript, it is only a summary of the included literature data and does not reflect the novelty of the research method or content. It is suggested that the author should be reconsider and revise the manuscript.

4. It is suggested that the author should be optimized the title of each part to make the logic of the results and the description of primary and secondary indicators clearer.

5. There are some details in the manuscript that need to revise, such as the literature in the risk assessment map should be arranged by year. In addition, the formatting of the manuscript is very terrible, especially for the layout of figures and tables, which need to be completely revised.

6. PLOS authors have the option to publish the peer review history of their article (what does this mean?). If published, this will include your full peer review and any attached files.

Reviewer #1: No

Reviewer #2: No

---

## [Author Response · Author response to Decision Letter 0]

1 Apr 2023

Additional Editor Comments:

The manuscript evaluated the adverse effects of two chemotherapy agents, suggesting that S-1 should be preferred in the treatment of gastric cancer.

There are some details in the manuscript that need to revise, such as the literature in the risk assessment map should be arranged by year. In addition, the formatting of the manuscript is very terrible, especially for the layout of figures and tables, which need to be completely revised. 

Response: We revised the literature, figures and tables according to your suggestion. 

Reviewer #1: Although the study was of great significance to explain the adverse events or reactions of 5-FU and S-1, the methods and reports of the manuscript did not meet the purpose of the study. My comments are as follows.

1.The review only included RCT and other types of studies such as non-randomized trials and case series reports were not collected. According to the PRISMA harms statement, these types of research advantages are important data sources.

Response: Due to the lack of scientific and reasonable control group, and easy to lead to misleading or deviation, it is not appropriate to conduct meta-analysis of non-randomized trials and case series.

2.Only adverse events were reported in the review. Adverse events need to analyze their causality with 5-FU and S-1. More importantly, in addition to the adverse events, these adverse reactions should also be reported and analyzed.

Response: Adverse events in our manuscript have included adverse reactions. Thanks for you comment. 

Reviewer #2: The manuscript evaluated the adverse effects of two chemotherapy agents, suggesting that S-1 should be preferred in the treatment of gastric cancer. It has certain significance to guide the clinical use of chemotherapy agents. But, some issues should be mentioned.

1.The retrieval dates of the include literature in the manuscript need to be updated to ensure the rigor of the manuscript and the accuracy of the data included.

Response: According to your suggestion, we retrieval the literature up to March 31st, 2023. However, we failed to find new literature that meet the inclusion criteria of our manuscript.

2.Has the PROSPERO registration been conducted for this Meta-analysis? If so, please provide the registration number.

Response: This meta-analysis is registered in the PROSPERO, and the registration number is CRD42023409814. Thanks for your suggestion. 

3.By reading the entire manuscript, it is only a summary of the included literature data and does not reflect the novelty of the research method or content. It is suggested that the author should be reconsider and revise the manuscript.

Response: Nursing can play an important role in controlling and alleviating adverse events caused by chemotherapy. This manuscript provides valuable reference for effectively controlling adverse events caused by chemotherapy and implementing effective nursing measures, which is an innovation of this manuscript. 

4.It is suggested that the author should be optimized the title of each part to make the logic of the results and the description of primary and secondary indicators clearer.

Response: It is revised. Thanks for you advised. 

5.There are some details in the manuscript that need to revise, such as the literature in the risk assessment map should be arranged by year. In addition, the formatting of the manuscript is very terrible, especially for the layout of figures and tables, which need to be completely revised.

Response: Thanks for your advise. The literature, figures and tables have revised.

---

## [Decision Letter · Decision Letter 1]

27 Jul 2023

PONE-D-23-03394R1S-1 and 5-Fluorouracil-related adverse events in patients with advanced gastric cancer: A meta-analysisPLOS ONE

Dear Dr. Chu,

Thank you for submitting your manuscript to PLOS ONE. After careful consideration, we feel that it has merit but does not fully meet PLOS ONE’s publication criteria as it currently stands. Therefore, we invite you to submit a revised version of the manuscript that addresses the points raised during the review process.

We look forward to receiving your revised manuscript.

Kind regards,

Sherief Ghozy, M.D.

Academic Editor

PLOS ONE

Reviewers' comments:

Reviewer's Responses to Questions

**Comments to the Author**

1. If the authors have adequately addressed your comments raised in a previous round of review and you feel that this manuscript is now acceptable for publication, you may indicate that here to bypass the “Comments to the Author” section, enter your conflict of interest statement in the “Confidential to Editor” section, and submit your "Accept" recommendation.

Reviewer #1: All comments have been addressed

Reviewer #2: All comments have been addressed

2. Is the manuscript technically sound, and do the data support the conclusions?

Reviewer #1: Partly

Reviewer #2: Yes

3. Has the statistical analysis been performed appropriately and rigorously? 

Reviewer #1: Yes

Reviewer #2: Yes

4. Have the authors made all data underlying the findings in their manuscript fully available?

Reviewer #1: Yes

Reviewer #2: Yes

5. Is the manuscript presented in an intelligible fashion and written in standard English?

Reviewer #1: Yes

Reviewer #2: Yes

6. Review Comments to the Author

Reviewer #1: Although the author has made revisions, the manuscript still has obvious shortcomings.

1. The authors have reported the results and methods used for assessing the risk of bias in individual studies. However, the risk of bias assessment should be considered separately for outcomes of benefit and harm.

2. For adverse events analysis, inclusion in randomized controlled trials is one-sided and insufficient.

3. The characteristics of the included studies remain underreported (Table 2). Some characteristics including dose, frequency, and course of treatment were not mentioned in the table.

4. Conjunctive therapy use of other drugs (Cisplatin, Paclitaxel) was also seen in the two groups in the included studies. The evaluation of the association of adverse events with interventions may be more complex. Also, it does not describe any assessment of possible causality.

Reviewer #2: The manuscript evaluated the adverse effects of two chemotherapy agents, suggesting that S-1 should be preferred in the treatment of gastric cancer. It has certain significance to guide the clinical use of chemotherapy agents. This work has been adequately revised. It could be accepted for publication.

7. PLOS authors have the option to publish the peer review history of their article (what does this mean?). If published, this will include your full peer review and any attached files.

Reviewer #1: No

Reviewer #2: **Yes: **Xiao Ma

---

## [Author Response · Author response to Decision Letter 1]

29 Jul 2023

Reviewer #1: 

Although the author has made revisions, the manuscript still has obvious shortcomings. 

1. The authors have reported the results and methods used for assessing the risk of bias in individual studies. However, the risk of bias assessment should be considered separately for outcomes of benefit and harm.

Response: 

In this work, the methods used to assessing the risk of bias in individual studies have been stated on page seven, lines 103 to 106, and the results of deviation assessment for individual studies have shown at Figs 2, 3 and 4. The purpose of bias evaluation is to test the quality of the included studies, which is used to confirm whether the reliability of the conclusion of meta-analysis is trustworthy. In addition, it is well known that the quality of included studies is positively related to the reliability of conclusions. The quality of the included studies in this meta-analysis is relatively high, so the conclusion of this work is highly reliable.

2. For adverse events analysis, inclusion in randomized controlled trials is one-sided and insufficient.

Response: 

Due to the lack of scientific and reasonable control group, other types of studies (such non-randomized trials, case reports and so forth) are likely lead to misdirection or bias and not suitable for conducting meta-analysis. 

3. The characteristics of the included studies remain underreported (Table 2). Some characteristics including dose, frequency, and course of treatment were not mentioned in the table.

Response: 

Dose, frequency and course of treatment have been added into Table 2. Thanks for your advice. 

4. Conjunctive therapy use of other drugs (Cisplatin, Paclitaxel) was also seen in the two groups in the included studies. The evaluation of the association of adverse events with interventions may be more complex. Also, it does not describe any assessment of possible causality. 

Response: 

Conjunctive therapy use of other drugs (Cisplatin, Paclitaxel) was also seen in the two groups in the included studies, which means there is comparability between S-1 and 5-FU groups. In addition, the aim of this study is to reveal the occurrence of adverse events caused by S-1 and 5-FU respectively. Therefore, the assessment of possible causality is another issue to be studied in the future. 

Reviewer #2: 

The manuscript evaluated the adverse effects of two chemotherapy agents, suggesting that S-1 should be preferred in the treatment of gastric cancer. It has certain significance to guide the clinical use of chemotherapy agents. This work has been adequately revised. It could be accepted for publication. 

Response: 

Thank you very much for your comments and support.

---

## [Editor Report · Decision Letter 2]

1 Aug 2023

S-1 and 5-Fluorouracil-related adverse events in patients with advanced gastric cancer: A meta-analysis

PONE-D-23-03394R2

Dear Dr. Chu,

We’re pleased to inform you that your manuscript has been judged scientifically suitable for publication and will be formally accepted for publication once it meets all outstanding technical requirements.

Kind regards,

Sherief Ghozy, M.D.

Academic Editor

PLOS ONE

---

## [Editor Report · Acceptance letter]

4 Aug 2023

PONE-D-23-03394R2 

S-1 and 5-Fluorouracil-related adverse events in patients with advanced gastric cancer: A meta-analysis 

Dear Dr. Chu:

I'm pleased to inform you that your manuscript has been deemed suitable for publication in PLOS ONE. Congratulations! Your manuscript is now with our production department. 

Kind regards, 

on behalf of

Dr. Sherief Ghozy 

Academic Editor

PLOS ONE